# Comparison of neonatal systemic and intracerebroventricular AAV9 gene therapy delivery demonstrating improved behavioral and phenotypic outcomes in a mouse model of Niemann-Pick disease, type C1

**Benjamin E. Epstein**[1], **Gabrielle M. Soden**[1], **Arturo A. Incao**[1], **Avani Mylvara**[1], **Jonathan Flynn**[2], **Fatih Ozsolak**[2], **William J. Pavan**[1]*

**1** Genomics, Development and Disease Section, Computational and Statistical Genomics Branch, National Human Genome Research Institute (NHGRI), National Institutes of Health, Bethesda, Maryland, United States of America, **2** Biologics Research Center, Novartis, San Diego, California, United States of America

* wjpavan@gmail.com

## Abstract

Niemann-Pick disease, type C (NPC), is an inherited fatal lysosomal storage disorder caused by a mutation in the *NPC1* or *NPC2* genes and characterized by impaired lysosomal cholesterol export. Previous studies have demonstrated that delivery of the *NPC1* gene to the central nervous system (CNS) via an adeno-associated virus (AAV) can substantially improve lifespan and mitigate signs of disease in *Npc1*-deficient mouse models of NPC. To determine the optimal parameters for an efficacious AAV-based gene therapy for NPC, we measured the survival and disease phenotypes of mice treated systemically as neonates or at weaning age, along with neonatal mice treated via intracerebroventricular (ICV) delivery, with a construct containing either a ubiquitous truncated EF1α promoter or a truncated *Mecp2* promoter. While all constructs and delivery methods resulted in improvement compared with baseline, mice treated as neonates survived significantly longer and experienced slower disease progression compared with those treated systemically at weaning age. Systemic delivery to neonates was capable of increasing survival and phenotypic improvement comparable to that of ICV delivery, and neonatal systemic and ICV delivery were both similarly capable of near-total Purkinje cell rescue. We also found no difference between a ubiquitous EF1α-derived promoter and an *Mecp2*-derived promoter. Ultimately, early treatment with maximal access to the CNS, whether via systemic or direct CNS delivery, is key to the efficacy of gene therapy in treating NPC.

**Data availability statement:** All relevant data are within the paper and its Supporting Information files.

**Funding:** This work was supported by the National Human Genome Research Institute's (NHGRI) Intramural Research Program (1ZIAHG000068-16) https://www.genome.gov/ (WJP). The funder had no role in study design, data collection and analysis, decision to publish, or preparation of the manuscript. This work was also funded by an NIH Cooperative Research and Development Agreement (CRADA) with Novartis Gene Therapies, Inc. https://www.novartis.com/ (CRADA #C-073-2023_A1) (WJP, FO).

**Competing interests:** I have read the journal's policy and the authors of this manuscript have the following competing interests: The NIH has filed patents on behalf of William J. Pavan on work related to NPC1 genes and the AAV gene therapy treatment of NPC1 (US Patent Publication Numbers 20180104289, 20210113635). Fatih Ozsolak and Jonathan Flynn are at present, or were during the time of their contribution to this manuscript, employed by Novartis Pharma AG and received salaries and own stock in Novartis as part of their remuneration for employment; they have no competing interests as regards consultancies, patents or products in development or currently marketed. This does not alter the authors' adherence to all the PLoS ONE policies on sharing data and materials.

## Introduction

Niemann-Pick disease, type C (NPC), is a rare, monogenic, fatal neurodegenerative disease affecting approximately 1:100,000 live births in the United States [1]. In approximately 95% of NPC individuals, the disease is caused by mutations in the *NPC1* gene, which encodes a lysosomal membrane-bound protein involved in the export of unesterified cholesterol from the lysosome [2,3]. The remaining ~5% of NPC individuals have mutations in *NPC2*, which encodes a soluble lysosomal protein thought to be involved in transfer of cholesterol to NPC1 for export [4–6]. Individuals with NPC experience progressive neurodegeneration, especially in the Purkinje cells of the cerebellum, as well as peripheral disease, with spleen and liver pathology being most prevalent [3]. Disease manifestations include cerebellar ataxia, cognitive impairment and developmental delay, respiratory dysfunction, and hepatosplenomegaly [1,7]. Diagnosis can occur at a broad range of ages, though earlier age of neurological symptom onset correlates with severity of disease [8,9]. There is no identifiable difference in clinical presentation between individuals with mutations in *NPC1* compared with mutations in *NPC2* [1]. Several small molecule therapies have progressed to or through Phase 3 clinical trials, including miglustat [10,11], Adrabetadex [12,13], Trappsol® Cyclo™ [14], arimoclomol [15], and *N*-acetyl-l-leucine [16], and two have been approved by the FDA [17]: Miplyffa (arimoclomol) [18] and Aqneursa (*N*-acetyl-l-leucine) [19]. AAV-based gene therapy, as an alternative to small molecule therapies, has been shown to significantly extend lifespan and slow signs of progression in mouse models of NPC via intravenous retro-orbital administration at weaning age [20,21], intracardiac delivery in neonates [22], and intracerebroventricular (ICV) delivery in neonates [23–25]. However, no direct comparison of different delivery methods has been performed to assess the qualitative and quantitative differences between them. A direct assessment of the significance of relative age of delivery is important to weigh potential clinical benefits of earlier treatment against the challenges of diagnosis at a younger age and the risks in treatment then. Further, the potential efficacy benefits of central nervous system (CNS) administration compared with systemic delivery must be assessed relative to the potential clinical challenges and risks. In this study, we compared systemic delivery in both neonates and weaning-age mice along with ICV delivery in neonates. We found that earlier delivery approaches were capable of substantial Purkinje cell rescue through systemic as well as direct ICV delivery methods, both showing consistent improvement in lifespan and disease progression rate, suggesting that earlier treatment results in both quantitative and qualitative improvement in our mouse model of NPC.

## Materials and methods

### Animals

Heterozygous *Npc1*$^{+/-}$ mice (BALB/cNctr-Npc1$^{m1N}$/J, Jackson Laboratory strain 003092) were crossed to obtain homozygous *Npc1*$^{-/-}$ mice and control *Npc1*$^{+/+}$ littermates. Humane endpoint criteria were predefined and approved by the NHGRI Animal Care and Use Committee. These criteria have been utilized in previous

studies and were selected to minimize suffering and distress in mice while maintaining a reproducible, rigorous survival metric. Mice were considered to have reached a humane endpoint and were therefore euthanized upon displaying at least two of the following signs, as previously described: loss of 30% of maximum weight, reluctance to move around the cage, repeated falling to the side while walking, and palpebral closure/dull rather than bright eyes [21]. The survival study lasted until the mice reached humane endpoint and were euthanized (n = 79 $Npc1^{-/-}$ mice) or, on very rare occasion, were unexpectedly found dead (n = 5 $Npc1^{-/-}$ mice).

Cause of death in mice is unknown as no necropsies were performed and every effort was made to euthanize at humane endpoint for tissue collection and further analysis. The use of analgesics or anesthetics does not ameliorate disease progression in $Npc1^{-/-}$ mice (which results from neuronal death) and adds confounding variability to the survival metric. Mice were therefore treated only with inert substances, such as eye ointment, and nail trims were proactively performed on $Npc1^{-/-}$ mice to minimize self-injury of eyes as motor function impairment progressed.

## Construction and production of AAV vectors

The AAV9-EF1α(s)-hNPC1 plasmid containing inverted terminal repeats (ITRs) from AAV serotype 2, a truncated EF1α promoter (EF1α(s)) for ubiquitous expression, the human $NPC1$ (hNPC1) cDNA, and a synthetic polyadenylation sequence was previously described [20]. The EF1α(s) promoter was replaced with a truncated murine $Mecp2$ promoter (designated "546") to generate the AAV9–546-hNPC1 plasmid. Recombinant AAVs were generated by triple transfection of HEK-293T cells using polyethylenimine. Viral particles were harvested from the cell pellet and culture medium 72 hr post transfection. Pellets were lysed with 0.1% Triton X-100, and the lysate was combined with AAV particles precipitated from culture medium with PEG-8000 as previously described [26]. Total lysate was clarified by centrifugation and 0.45 µm filtration. Clarified stock was purified with POROS™ CaptureSelect™ AAV9 Affinity Resin (Thermo Scientific) followed by iodixanol gradient centrifugation. AAV particles were concentrated using Ultra-15 100 kDa Centrifugal Filter Unit (Amicon) and exchanged into suspension buffer (Tris-HCL 20 mM, $MgCl_2$ 1 mM, NaCl 200 mM, 0.05% Pluronic® F-68, pH 8.0) followed by sterile filtration with a 0.2 µm syringe filter (AcroDisc). AAV titers were determined by RT-qPCR on a LightCycler® 480 (Roche) using ITR-specific PCR primers [27].

## Delivery of AAV

$Npc1^{-/-}$ mice received an injection of an AAV vector (AAV9-EF1α(s)-hNPC1 or AAV9–546-hNPC1 diluted in PBS + 0.001% Pluronic® F-68) or 0.9% saline via one of the following methods. For the post-weaning age systemic (RO) cohort: Mice aged 31–34 d were anesthetized via isoflurane inhalation and administered a 50-µL retro-orbital injection of $1.21 \times 10^{12}$ genome copies (GC) of the desired vector or saline. For the neonatal systemic facial vein (FV) cohort: Mice aged 1 d were anesthetized on ice for 60 s and administered a 30-µL injection of $1.0 \times 10^{11}$ GC or saline into the facial vein using a 3/10 cc syringe as previously described [28]. For the neonatal ICV cohort: Mice aged 1 d were anesthetized on ice for 60 s and administered a 5-µL injection of $1.0 \times 10^{10}$ GC or saline into the right lateral ventricle using a Hamilton syringe fitted with a 32-gauge stainless steel needle as previously described [29]. Coordinates for injection were halfway between bregma and lambda and 1 mm lateral to lambda at a depth of 2 mm [29]. For neonatal injections, vectors were delivered to entire litters of mice at P1 and genotyping was performed at 2 wk. Litters were injected until at least three male and three female $Npc1^{-/-}$ mice received each treatment (vector or saline). Additional $Npc1^{-/-}$ mice injected as a result beyond that number were enrolled in the study as well.

## Phenotype assessments

Mice were weighed weekly beginning at 6 wk of age (when weights of WT and untreated $Npc1^{-/-}$ mice begin to diverge [20]) and as frequently as daily as they progressed toward humane end point. Mice were assessed every 2 wk beginning at 6 wk, as well as at 9 wk, for disease phenotypes of grooming, gait, kyphosis, ledge balance, and hindlimb clasp using

the quantitative phenotypic scoring system for assessing NPC mouse models described previously, with higher scores on each test indicating greater disease progression [30]. Mice were also assessed on these weeks for motor function on a balance beam test described previously [31,32]. Mice traversed a 4′ wooden beam (diameter 18″), and their hind-foot slips off the beam were counted. Mice were given three attempts to cross the beam without falling. Mice were not penalized for dismounts from the beam unrelated to disease phenotype (e.g., distraction). All phenotype assessments were performed by two observers blinded to the treatment status of the mice and the results averaged.

## Euthanasia and tissue extraction

Mice were deeply anesthetized via an intraperitoneal injection of avertin (500 mg/kg) and transcardially perfused with 30 mL of PBS. The right half of the cerebrum and a piece of liver were immersed in 10 µL of DNA/RNA Shield (Zymo Research) per mg of tissue. Tissues were immediately snap frozen in a bath of dry ice and ethanol and stored at −80°C. The left half of the brain was immersed in 4% PFA and placed at 4°C for 48 hr. Tissues were then washed three times with PBS for 10 minutes per wash. Tissues were stored in PBS + 0.01% sodium azide until preparation for sectioning.

## Tissue sectioning and staining

Fixed tissue samples were immersed in 30% sucrose in PBS until the tissue sank. Tissues were then placed into Seal'N Freeze Cryotray cryomolds (Klarex Health), covered with OCT compound, and frozen in a bath of dry ice and ethanol followed by storage at −80°C.

Brain tissue was acquired at end stage for immunohistochemical analysis. Histoserv, Inc. (Germantown, MD) performed sectioning of frozen brain tissue. Sagittal brain sections (10 µm) were collected on slides. Samples were incubated with 1.6% $H_2O_2$ for 10 min, washed in PBS followed by 0.25% Triton X-100 in PBS (PBST), blocked for 1 hour at room temperature with PBST/normal goat serum, and incubated in primary antibody at 4°C overnight. For immunofluorescence imaging, after washing in PBST again, samples were incubated with secondary antibody for 30 min at room temperature. Finally, samples were coverslipped with ProLong™ Gold Antifade Mountant or ProLong™ Gold Antifade Mountant with DAPI (Invitrogen) for immunofluorescent imaging.

Primary antibodies included: α-Calbindin (Abcam ab229915), α-IBA1 (Wako Chemicals 019–19741), α-GFAP (Sigma G3893). Secondary fluorescent antibodies included: Goat anti-mouse IgG AlexaFluor 488 (ThermoFisher Invitrogen A11029), Goat anti-rabbit IgG AlexaFluor 488 (ThermoFisher Invitrogen A11034), Goat anti-rabbit IgG AlexaFluor 594 (ThermoFisher Invitrogen A11037), Goat anti-rat IgG AlexFluor 594 (A11007).

## Copy number analysis

Tissue samples were homogenized in DNA/RNA Shield (Zymo Research) with a BeadBug™ 6 homogenizer (Benchmark Scientific). Genomic DNA was extracted from tissue homogenate with a *Quick*-DNA/RNA Miniprep Plus Kit (Zymo Research). Vector copy number was assessed via droplet digital PCR (ddPCR). Samples were prepared with ddPCR copy number assays containing premixed primers and TaqMan probes for hNPC1 labeled with FAM (Bio-Rad, cat. #10042958, assay ID: dCNS361140976) and GAPDH labeled with HEX (Bio-Rad, cat. #10042961, assay ID: dMmuCNS300520369). Samples were combined with ddPCR Supermix for Probes (Bio-Rad) and run on a QX200 Droplet Digital PCR System (Bio-Rad) with Copy Number Variation (CNV) settings chosen using thermal cycling conditions described by the manufacturer for use with ddPCR Supermix for Probes. Copy number was calculated using QuantaSoft Software v. 1.7.4 (Bio-Rad).

## Statistical analysis

All statistical analyses were performed using GraphPad Prism version 10.1.2 for Windows (GraphPad Software, San Diego, California USA, www.graphpad.com). Values and errors reported are means and standard deviations unless

otherwise indicated. Survival curves were compared pairwise via log-rank Mantel-Cox test with Bonferroni correction applied manually for multiple comparison correction. Correlation r values reported are Pearson correlation coefficients. Slopes of linear regressions were compared pairwise via ANCOVA with Bonferonni correction applied manually for multiple comparison correction. Balance beam traversal success rates were compared with Fisher's exact test with Bonferroni correction applied manually for multiple comparison correction. Two-way ANOVAs were performed with delivery route/age and promoter as factors and were followed by a Holm-Šídák multiple comparisons test for main column effects. One-way ANOVAs were followed by a Holm-Šídák multiple comparisons test. Multiplicity adjusted *P* values are reported to account for multiple comparisons. For statistical tests with Bonferroni correction applied manually, alphas for statistical significance of each comparison are listed along with *P* values. Ratios of liver and cerebrum copy numbers were log-transformed prior to statistical analysis.

## Results

### Different delivery routes result in differential access to brain and liver tissue

AAV vectors were prepared containing a human *NPC1* (hNPC1) cassette driven by one of two different promoters: a truncated form of the ubiquitous elongation factor-1 alpha promoter (EF1α(s)), or a small *Mecp2*-derived promoter ("546"). Table 1 outlines the study design in which cohorts of at least six *Npc1*$^{-/-}$ mice (≥ 3 male, ≥ 3 female) were treated with one of six combinations of vector (AAV9–546-hNPC1, AAV9-EF1α(s)-hNPC1) and delivery route (P31-34 retro-orbital (RO), P1 neonatal facial vein (FV), P1 neonatal intracerebroventricular (ICV)). Control cohorts of saline-injected *Npc1*$^{-/-}$ and uninjected *Npc1*$^{+/+}$ mice were also included. To assess the transduction efficiency and biodistribution of each of these vectors and delivery routes, cerebrum, cerebellum, and liver tissue samples collected after mice reached end stage were assessed by droplet digital PCR (ddPCR). Assessment of the cerebrum showed significantly higher copy numbers in ICV mice (3.3 ± 2.6) compared with FV (1.5 ± 2.3, *P* = 0.0295) or RO mice (0.4 ± 0.2, *P* = 0.0009) (Fig 1A). RO mice had relatively tightly clustered copy number values (range = 0.23–0.84, CV = 45.1%). ICV mice had greater variability than RO mice but clustered significantly higher (range = 0.85–9.84, CV = 79.32%), indicating greater transduction of the brain. FV mice, however, had greater variability (range = 0.13–9.08, CV = 148.2%). Analysis of the FV cohort reveals that mice clustered into two subpopulations, with some mice having copy number values clustering near those of RO mice and others having values 10- to 40-fold higher, resembling those of ICV mice. No significant difference was found in cerebrum CNV between cohorts injected with vector containing the EF1α(s) promoter and the 546 promoter (S1A Fig).

Systemic treatments (RO and FV) also have ready access to the liver. ICV delivery is largely restricted to the CNS, though leakage into the periphery from the cerebrospinal fluid after AAV9 administration has been shown to occur detectably [33]. As liver pathology is a common clinical sign of NPC, we assessed gene copy number in the liver as well

**Table 1. Treatment cohorts.**

| Genotype | Vector | Age | Delivery route | Dose (total GC) | Cohort size |
|---|---|---|---|---|---|
| *Npc1*$^{+/+}$ | N/A | N/A | N/A | N/A | 30 |
| *Npc1*$^{-/-}$ | None (saline) | P31-34 | RO | N/A | 16 |
| *Npc1*$^{-/-}$ | AAV9–546-hNPC1 | P31-34 | RO | $1.21 \times 10^{12}$ | 12 |
| *Npc1*$^{-/-}$ | AAV9-EF1α(s)-hNPC1 | P31-34 | RO | $1.21 \times 10^{12}$ | 14 |
| *Npc1*$^{-/-}$ | AAV9–546-hNPC1 | P1 | FV | $1.00 \times 10^{11}$ | 8 |
| *Npc1*$^{-/-}$ | AAV9-EF1α(s)-hNPC1 | P1 | FV | $1.00 \times 10^{11}$ | 12 |
| *Npc1*$^{-/-}$ | AAV9–546-hNPC1 | P1 | ICV | $1.00 \times 10^{10}$ | 11 |
| *Npc1*$^{-/-}$ | AAV9-EF1α(s)-hNPC1 | P1 | ICV | $1.00 \times 10^{10}$ | 11 |

WT, wild-type *Npc1*$^{+/+}$; RO, retro-orbital injection; FV, facial vein injection; ICV, intracerebroventricular injection; P, post-natal day; GC, genome copies.

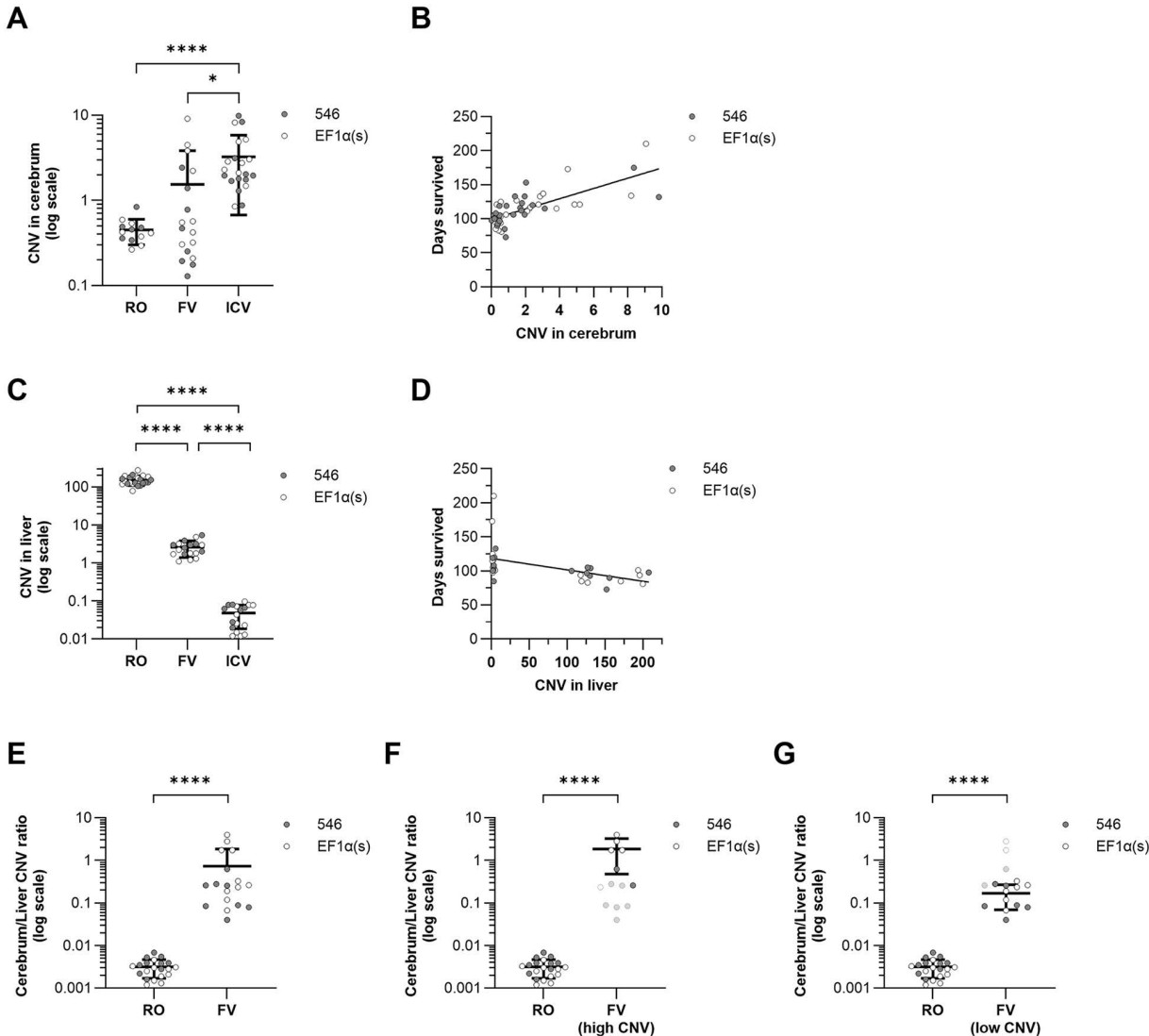

**Fig 1. Copy numbers in the cerebrum correspond to improved survival rates with higher end-stage cerebrum:liver copy number ratios found in systemic delivery in neonates than in weaning-age mice.** Tissue samples were analyzed via ddPCR for copy number of delivered hNPC1 DNA and compared via one-way ANOVA with multiple comparisons corrected by Holm-Šídák method unless otherwise indicated. **(A)** Copy number in the cerebrum. **(B)** Correlation between cerebrum copy number and survival time. **(C)** Copy number in the liver compared by t test. **(D)** Correlation between liver copy number and survival time. **(E)** Ratio of copy number in cerebrum to liver was assessed for efficiency of delivery to the brain. Ratios were log transformed prior to statistical analysis. **(F)** Cerebrum to liver copy number ratio for FV mice with high copy number in the cerebrum (CNV ≥ 1). **(G)** Cerebrum to liver copy number ratio for FV mice with high copy number in the cerebrum (CNV < 1). *$P < 0.05$, **$P < 0.01$, ***$P < 0.001$, ****$P < 0.0001$.

(Fig 1C). Copy numbers were 40- to 70-fold higher in RO than FV mice and more than 3000-fold higher in RO mice than in ICV mice (RO: 152.5 ± 43.62; FV: 2.6 ± 1.21; ICV: 0.048 ± 0.029; RO vs. FV: $P < 0.0001$; RO vs. ICV: $P < 0.0001$; FV vs. ICV: $P < 0.0001$). No significant difference was found in liver CNV between cohorts injected with vector containing the EF1α(s) promoter and the 546 promoter (S1B Fig).

The effective biodistribution of different routes of delivery can be partly assessed by comparing relative copy numbers in the cerebrum and the liver. The ratio of cerebrum copy number to liver copy number was collectively more than 200-fold higher in

FV mice than in RO mice (FV: 0.73±1.1, RO: 0.0032±0.0015, $P<0.0001$) (Fig 1E), indicating a greater proportion of total vector in the brain at end stage in FV mice than in RO mice. The cerebrum to liver ratio was significantly higher in FV mice with higher copy numbers in the cerebrum than those with lower copy numbers in the cerebrum, though both were still significantly greater than that of RO mice (high-CNV FV: 1.9±1.4, $P<0.0001$; low-CNV FV: 0.17±0.10, $P<0.0001$) (Fig 1F–G).

## Neonatal treatment with AAV-hNPC1 improves survival more than weaning-age treatment

Analysis of survival by two-way ANOVA found delivery route, but not promoter choice (S1C Fig), to be a statistically significant source of variation in survival time (33.41% of total variation, $P<0.0001$). All three delivery routes significantly increased median survival rate over saline-treated mice (saline: 10.6 wk) (Fig 2A). Further, both the neonatal FV treatment and neonatal ICV treatment significantly improved survival rate over weaning-age RO treatment (FV: 16.1 wk, ICV: 17.3 wk, RO: 13.5 wk) and were not significantly different from each other (Fig 2A). There was a significant correlation between copy number in the cerebrum and survival time (r=0.73, $P<0.0001$) (Fig 1B), suggesting that a primary driver of phenotypic improvement is successful access to the brain. Copy number in the liver, however, did not correlate with increased survival (r= −0.50, $P=0.003$) (Fig 1D).

As described, some FV mice had cerebrum copy numbers comparable to those of ICV mice, while some had cerebrum copy numbers similar to those of RO mice (Fig 1A). Survival of ICV mice was compared with FV mice having similar copy numbers (CNV≥1), and no significant difference was found between the median survival time of the two populations (high-CNV FV: 18.1 wk, ICV: 16.9 wk, $P=0.0886$) (Fig 2B). However, a comparison of the FV mice having lower CNV (< 1) with RO mice found that low-CNV FV mice had longer median survival times than RO mice (low-CNV FV: 14.9, RO: 13.4, $P=0.003$, α=0.01) and a longer maximum survival time (low-CNV FV: 17.9 wk, RO: 15.6 wk), with 3/12 low-CNV FV mice living longer than all 23 RO mice (Fig 2C).

## Neonatal treatment with AAV-hNPC1 slows progressive weight loss

Saline-treated mice began to reach humane end point as early as 10 wk; phenotype assessments are therefore reported both over the lifespan of the mice as well as at 9 wk, the latest time point that includes every cohort in full. A primary indicator of decline in $Npc1^{-/-}$ mice is progressive weight loss. Analysis of weight change between 6 wk and 9 wk again found delivery route, but not promoter choice, to be a significant source of variation between cohorts (34.24%, $P<0.0001$); however, promoter and delivery route interaction was also a significant source of variation, though smaller in magnitude (11.60%, $P=0.002$) (S1D Fig). At 9 wk of age, saline-injected mice lost a mean of 16.8%±8.7% of their weight at 6 wk (Fig 3A). In contrast, treatment via all three delivery routes mitigated weight loss in this interval (RO: −2.0%±6.8%, FV: +3.9%±5.2%, ICV: +8.2%±4.9%, $P<0.0001$ for all vs. saline). Further, both FV and ICV treatment significantly improved weight change over RO treatment (RO vs. FV: $P=0.004$, RO vs. ICV: $P<0.0001$), and ICV treatment improved weight change over FV treatment ($P=0.03$). However, the low-CNV and high-CNV subsets of FV mice each experienced weight change respectively more similar to RO and ICV groups. Low-CNV FV mice did not significantly improve weight change over RO treatment (low-CNV FV: +2.0%±4.7%, RO: −2.0%±6.8%, $P=0.0564$) (Fig 3D), and high-CNV FV mice were not significantly different from ICV mice (high-CNV FV: +8.2%±3.7%, ICV: +8.2%±4.9%, $P>0.9957$) (Fig 3C).

Treatments that improve lifespan in $Npc1^{-/-}$ mice may increase the time interval until a mouse reaches its peak weight, the interval of declining weight between reaching peak weight and humane end point, or both. Saline-injected mice reach their peak weight at a mean age of 6.5 wk (± 0.5) before weight loss begins (Fig 3B). While RO treatment did not significantly change time to reach peak weight (7.3±0.7 wk, $P=0.159$), both neonatal FV treatment (9.1±3.0 wk, $P=0.0002$) and neonatal ICV treatment (10.3±2.4 wk, $P<0.0001$) significantly increased this time. In contrast, while saline-injected mice only survived for a mean of 4.1±0.7 additional weeks once weight loss began, all three treatments, including RO treatment, significantly slowed the rate of weight loss and increased post-peak weight survival time (RO: 6.1±1.6 wk, $P=0.005$; FV: 7.8±2.3 wk, $P<0.0001$; ICV: 7.4±2.0 wk, $P<0.0001$).

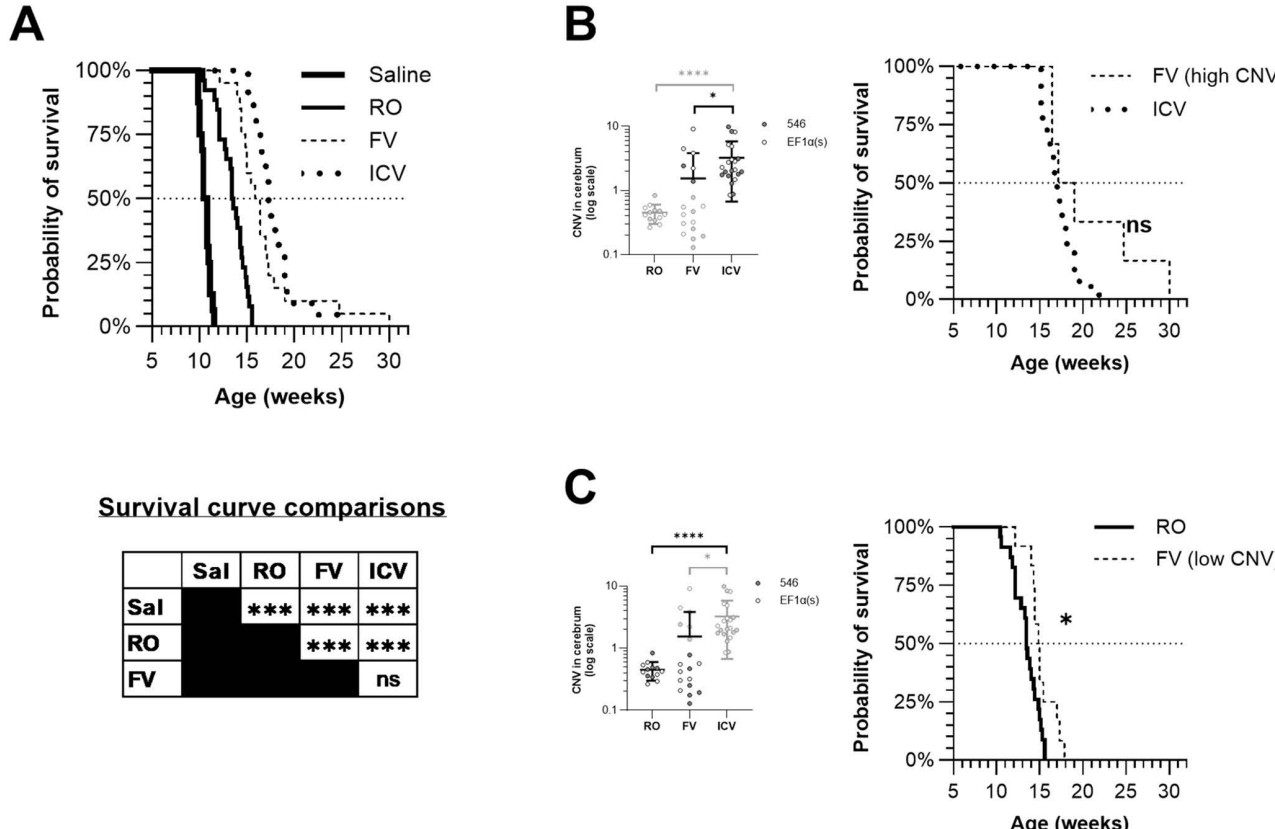

**Fig 2. Earlier treatment with AAV-hNPC1 improves survival as a function of cerebrum CNV.** Mice treated with vectors containing either the EF1α(s) or the 546 promoter are grouped together for each route of administration. **(A)** Kaplan-Meier survival curve showing survival rate of treated and untreated cohorts. No mice in the $Npc1^{+/+}$ cohort died over the course of the study (data not shown). α=0.0083, *$P$<0.0083, **$P$<0.0017, ***$P$<0.00017, ns: not significant. **(B)** Cerebral copy number of RO, FV, and ICV cohorts with high-CNV FV mice and ICV mice highlighted. Symbols in dark outlines are represented in the Kaplan-Meier survival curve; symbols in light grey outlines are not.(C) Cerebral copy number of RO, FV, and ICV cohorts with low-CNV FV mice and RO mice highlighted. Symbols in dark outlines are represented in the Kaplan-Meier survival curve; symbols in light grey outlines are not. α=0.025, *$P$<0.025.

## Neonatal treatment with AAV-hNPC1 slows phenotype progression

To assess the impact of different treatments on disease phenotypes, mice were assessed via a standard NPC phenotype test and a balance beam test. The phenotype test consists of observing five different disease-associated phenotypes, scoring each one between 0 (unaffected) and 3 (severe), and combining the results into a single composite score. For the balance beam test, mice traverse a wooden balance beam and are assessed for the ability to successfully traverse the entire beam as well as the number of hind-foot slips off the beam if they succeed. Loss of motor function leads to an increase in foot slips on the beam. After significant progression, mice lose the ability to traverse the beam without falling off.

At 9 wk, all vector-treated mice had significantly lower composite phenotype scores than saline-injected mice (Fig 4A). Additionally, ICV mice had significantly lower scores than RO mice (RO: 4.2±1.3, ICV: 3.1±1.1, $P$=0.0085). However, though FV mice were not statistically distinguishable from either RO or ICV mice, the high-CNV FV subset had significantly lower scores than RO mice (high-CNV FV: 2.7±0.7, $P$=0.0116) (Fig 4C) and were indistinguishable from ICV mice ($P$=0.4311). Low-CNV FV mice appeared similar to the larger cohort (Fig 4D). From 6 wk to 9 wk, all vector-treated mice

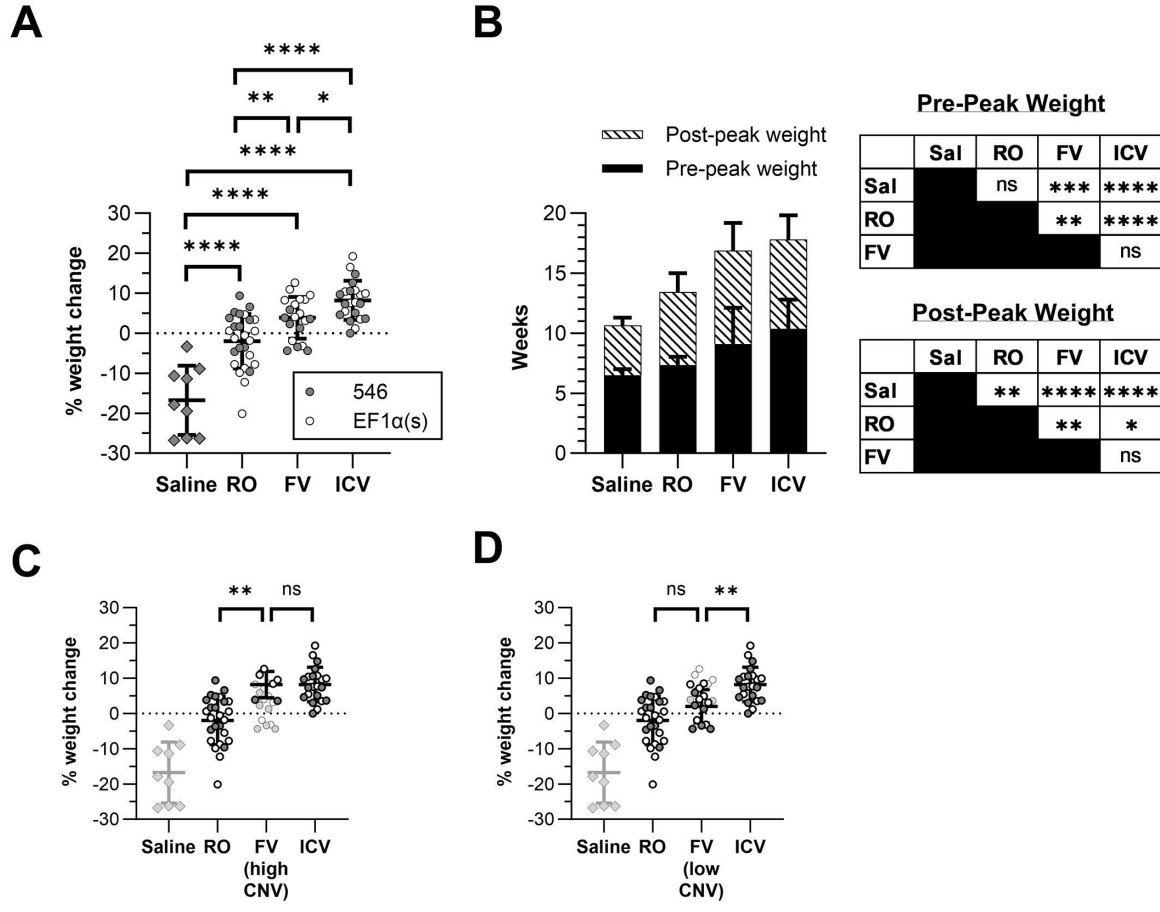

**Fig 3. Treatment with AAV-hNPC1 increases time until weight loss begins and slows rate of weight loss.** Groups compared by one-way ANOVA with multiple comparisons corrected by Holm-Šídák method unless otherwise indicated. **(A)** Percent increase or decrease in weight at 9 wk of age compared with weight at 6 wk. **(B)** Comparison of lifespan before weight loss begins (when peak weight is reached) and from the week of peak weight until humane end point. Data compared by two-way ANOVA with multiple comparisons corrected by Holm-Šídák method. Statistical significance for both phases is displayed in the table. **(C)** Percent weight change at 9 wk of age compared with 6 wk for FV mice with high CNV ($\geq$1). **(D)** Percent weight change at 9 wk of age compared with 6 wk for FV mice with low CNV in the brain (<1). *$P<0.05$, **$P<0.01$, ***$P<0.001$, ****$P<0.0001$, ns: not significant. Symbols in dark outlines are represented in the statistical analysis; symbols in light grey outlines are not.

also had a significantly slower rate of increase in phenotype score than saline-injected mice (Fig 4B). No significant difference was found in composite phenotype score between cohorts injected with vector containing the EF1α(s) promoter and the 546 promoter (S1E Fig).

The balance beam assay provides an objective and quantitative metric for the progression of motor function degeneration. Analysis of foot slip data by two-way ANOVA again found the delivery route, but not promoter choice (S1F Fig), to be a significant source of variation between cohorts, and it accounted for the largest percent of the variation of any metric (57.33%, $P<0.0001$). At 9 wk, only 25% of saline-treated successfully traversed the beam (Fig 5B). All gene therapy treatments increased the rate of traversal success at 9 wk relative to saline-treated mice, with 77% of RO mice (20/26, $P=0.0014$, $\alpha=0.0083$), 95% of FV mice (19/20, $P<0.0001$, $\alpha=0.0083$) and 100% of ICV mice (22/22, $P<0.0001$, $\alpha=0.0083$) successfully crossing the beam. Among the mice that successfully crossed the beam at this time point, every vector-treated group had fewer foot slips than saline-injected mice (saline: 73.8±9.8; RO: 59.8±7.6; FV: 35.7±23.5; ICV: 14.3±13.0) (Fig 5A). Further, all treatments significantly differed from each other, with FV mice slipping less than RO mice

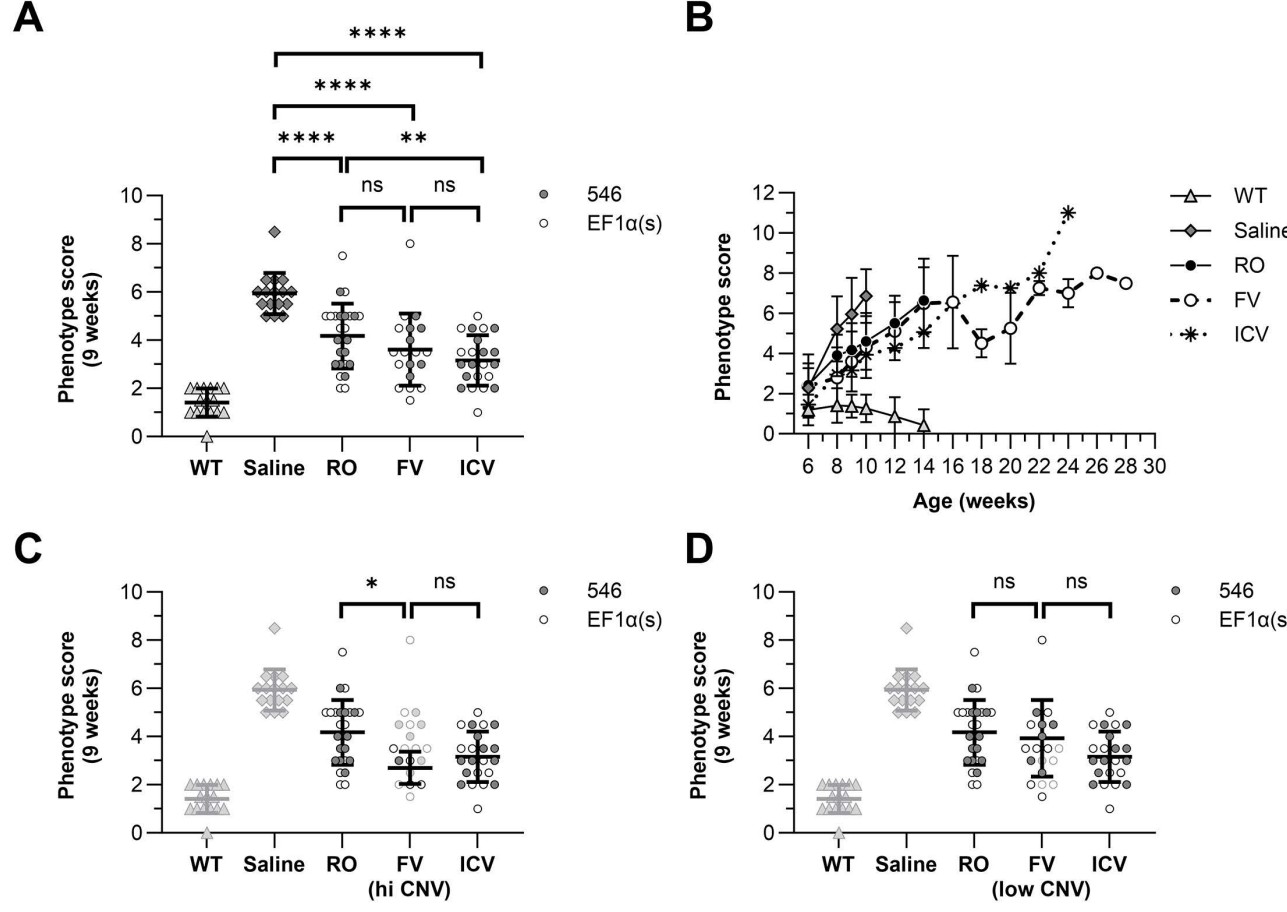

**Fig 4. Neonatal treatment with AAV-hNPC1 improves disease phenotype progression over later treatment.** Mice were scored between 0 and 3 on the five elements of the NPC1 phenotype assay and scores were combined into a composite score. Scores were compared by one-way ANOVA with multiple comparisons corrected by Holm-Šídák method. **(A)** Composite phenotype scores at 9 wk. **(B)** Composite phenotype scores over time. As mice reach humane end point, sample size decreases at later time points in each group. **(C)** Composite phenotype scores at 9 wk for FV mice with high CNV in the brain compared with RO and ICV mice. **(D)** Composite phenotype scores at 9 wk for FV mice with low CNV in the brain compared with RO and ICV mice. *$P<0.05$, ****$P<0.0001$ Symbols in dark outlines are represented in the statistical analysis; symbols in light grey outlines are not.

($P=0.0002$) and ICV mice slipping less than either FV or RO mice ($P<0.0001$). FV mice had the greatest variance in foot slips that only partly appeared related to vector delivery success. While seven of the eight mice that scored poorly with > 40 foot slips at 9 wk did have lower copy numbers in the brain (cerebrum CNV < 1), mice that scored better with < 40 foot slips had a large variance in cerebrum CNV between 0–9.1, and the overall correlation between foot slips and cerebrum CNV at 9 wk was not significant ($r= -0.45$, $P=0.068$). Even so, unlike the full cohort, high-CNV FV mice were not statistically significantly different from ICV mice (high-CNV FV: 24.4 ± 18.0, $P=0.08$) (Fig 5C), while low-CNV FV mice were distinct from both RO and ICV mice, similar to the larger cohort (low-CNV FV: 39.8 ± 24.5, low-CNV FV vs. RO: $P=0.0007$, low-CNV FV vs. ICV: $P<0.0001$) (Fig 5D).

In addition to performing worse on the balance beam at 9 wk, the different cohorts of mice declined in proficiency at different rates. Between 6 wk and 9 wk, linear regression analysis of foot slips over time shows that saline-injected mice progressed the fastest (slope = 22.8 ± 1.3, $R^2=0.92$), whereas each other treatment progressed significantly slower than saline-injected mice (RO: 17.0 ± 1.2, $R^2=0.74$; FV: 9.9 ± 2.0, $R^2=0.32$; ICV: 4.0 ± 0.8, $R^2=0.27$). Further, slopes of both FV

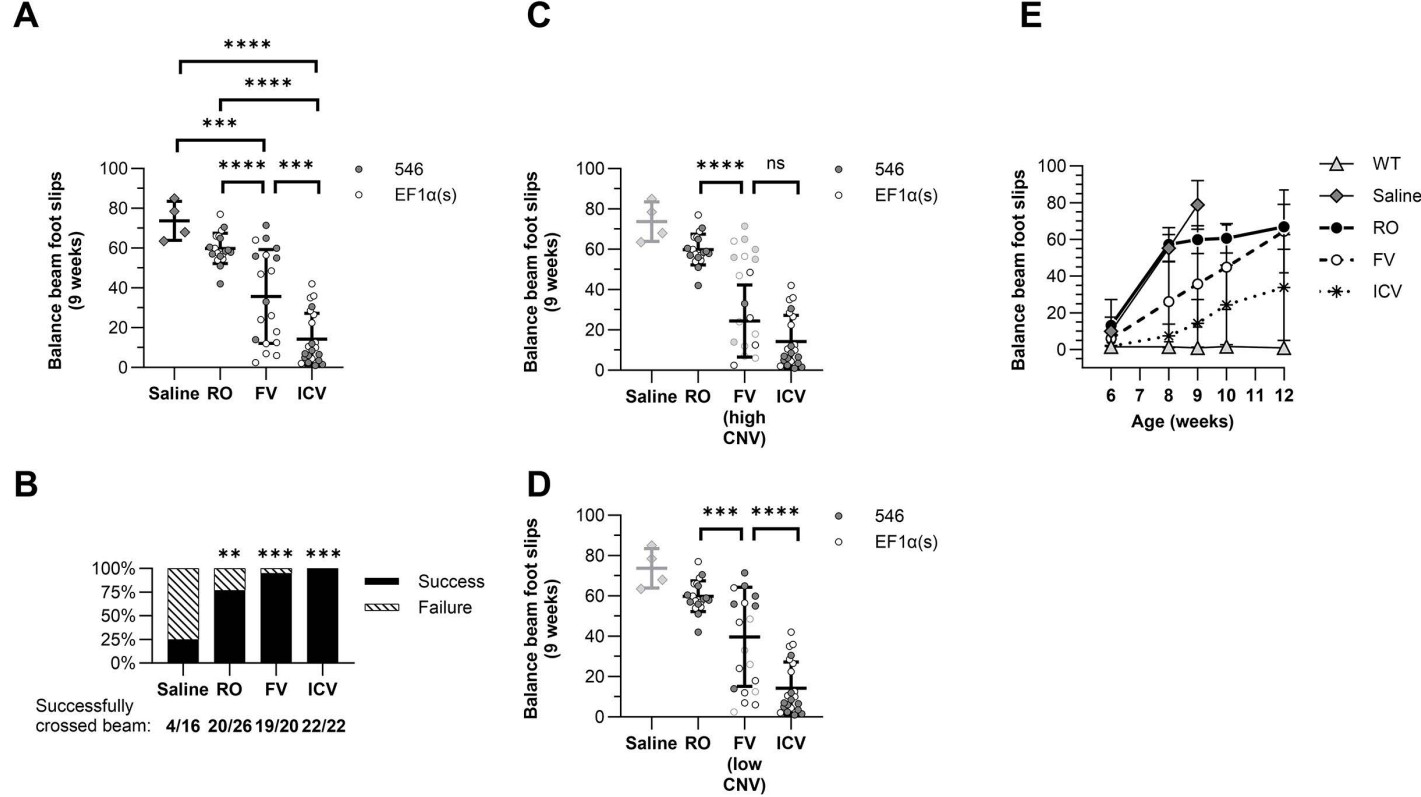

**Fig 5. Treatment with AAV-hNPC1 has greatest impact on motor coordination in the earliest groups treated.** Foot slips while traversing the balance beam were counted by two observers and averaged. Mice that did not successfully traverse the beam without falling were not included in this analysis due to lack of data on foot slips committed in complete traversal. Groups were compared by one-way ANOVA with multiple comparisons corrected by Holm-Šídák method unless otherwise indicated. **(A)** Foot slips at 9 wk of age. ***$P < 0.001$, ****$P < 0.0001$. **(B)** Number of mice successfully traversing the beam from each cohort compared by Fisher's exact test. Statistical significance is indicated in comparison to saline cohort; all other comparisons were not statistically significant. $\alpha = 0.0083$, **$P < 0.0017$, ***$P < 0.00017$. **(C)** Foot slips at 9 wk of age of FV mice with high CNV in the brain compared to RO and ICV groups. **(D)** Foot slips at 9 wk of age of FV mice with low CNV in the brain compared to RO and ICV groups. **(E)** Foot slips on the balance beam assay over time between 6 and 12 wk. All mice in the saline cohort had either reached humane end point or could not traverse the balance beam successfully by 10 wk.

and ICV cohorts were significantly smaller than those of RO mice (FV: $P = 0.0019$; ICV: $P < 0.0001$; $\alpha = 0.0083$), showing slower decline with earlier treatments (Fig 5E).

## Neonatal treatment with AAV-hNPC1 preserves Purkinje cells and reduces CNS pathology

Sagittal sections of brains for all mouse cohorts were taken from end-stage samples to assess long-term efficacy of treatment and assessed for signs of pathology. Anti-calbindin staining was performed to identify surviving Purkinje cells in the cerebellum and subjective assessments are as follows. Saline-injected mice had few surviving Purkinje cells at end stage, all in posterior lobules (Fig 6A), and RO mice cerebella were similar (Fig 6B). FV mice again divided into two groups, with those receiving a higher effective dose seeing near complete preservation of Purkinje cells (n=3) (Fig 6C), and those receiving a lower effective dose resembling saline and RO mice with Purkinje cell survival only in posterior lobes (n=5) (Fig 6D). ICV mice uniformly had complete preservation of Purkinje cells (Fig 6E). Level of preservation predicted a trend toward higher survival time, with median survival of the better preserved FV mice trending higher than median survival time of the less preserved FV mice, and strikingly, also trending higher than that of the ICV group, though the difference in pairwise

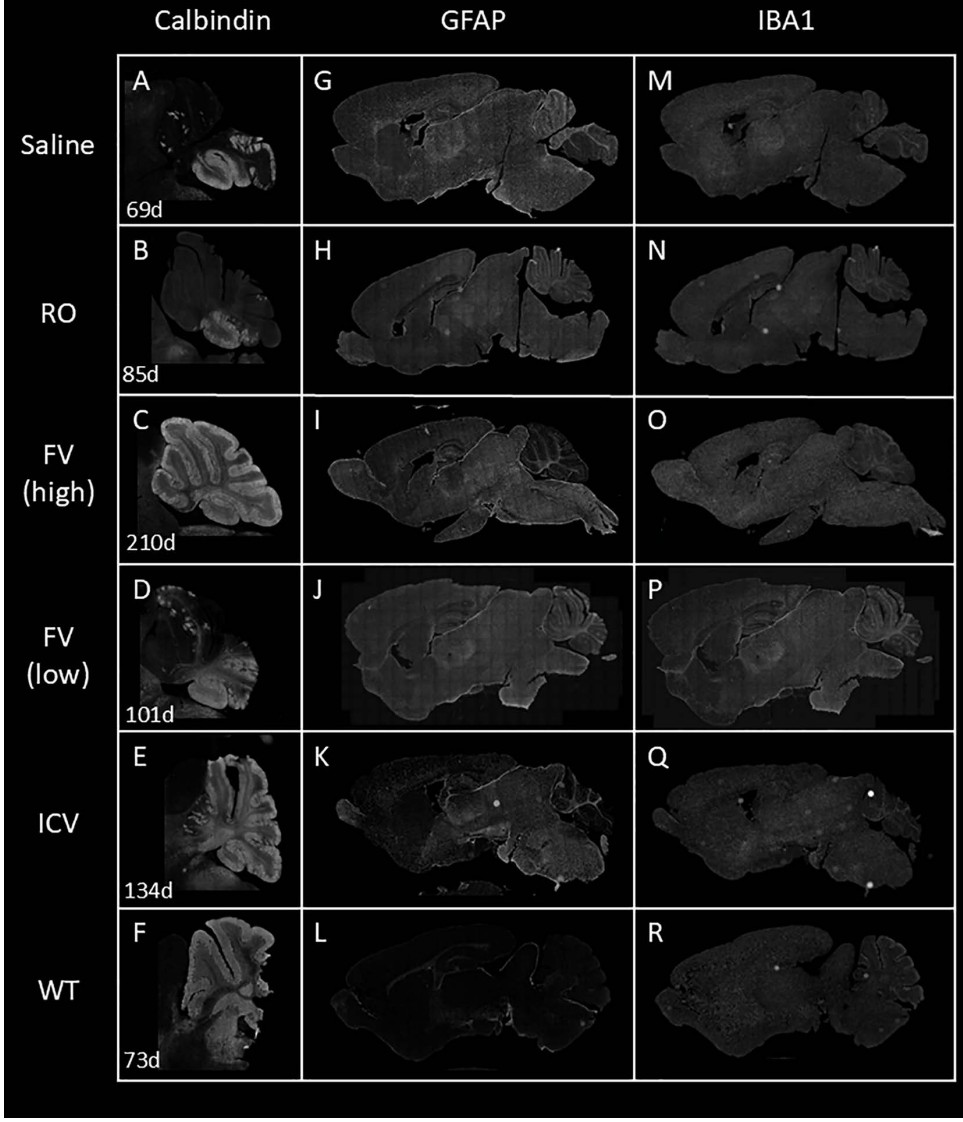

**Fig 6. Treatment with AAV9-EF1α(s)-hNPC1 protects Purkinje cells and reduces CNS pathology in treated mice at end stage.** Sagittal sections were stained with anti-calbindin (Purkinje cells in the cerebellum) **(A–F)**, anti-GFAP (astrocytic marker) **(G–L)**, or anti-IBA1 (microglial marker) **(M–R)**. All mice except for WT mice were sacrificed at end stage at ages shown **(A–F)**.

comparisons across all three groups was not statistically significant (preserved: 23.2 wk; unpreserved: 15.4 wk; ICV: 17.29 wk; preserved vs. unpreserved: $P=0.08$, $α=0.001$; preserved vs. ICV: $P=0.08$, $α=0.001$) (Fig 2B). No significant difference was observed in images from cohorts injected with vector containing the EF1α(s) promoter and the 546 promoter.

Sections were also stained with antibodies against IBA1 to assess microglial activation (Fig 6M–R) and GFAP (Fig 6G–L) to visualize astrocytosis. While RO cohorts showed a minimal reduction in staining (Fig 6H), high-CNV FV cohorts saw a greater reduction in cerebellar GFAP staining (Fig 6I). ICV cohorts generally showed a modest reduction in IBA1 staining and a clear reduction in GFAP staining, indicating reduced astrocytosis in transduced regions (Fig 6K, Q). The greatest reduction in staining appeared in the cerebral cortex, the anterior olfactory zone, the subventricular zone, and the cerebellum, and improvement increased with higher delivered copy numbers.

## Discussion

Our findings demonstrate that AAV gene therapy for NPC using AAV9 can be improved both by earlier administration and by direct CNS delivery. Further, we found no significant difference between the efficacy of a minimal ubiquitous EF1α-derived promoter (EF1α(s)) and an *Mecp2*-derived promoter (546).

We demonstrated a marked improvement in survival and phenotype progression in both neonatal FV- and ICV-administered AAV gene therapy groups. While previous studies have separately shown the efficacy of systemic gene therapy at weaning age [20,21] and ICV gene therapy in neonates [23–25], our study is the first to directly compare the efficacies of the two approaches. These approaches also assess two variables simultaneously: route of administration (systemic vs. direct CNS delivery) and age of administration (neonate vs. weaning age). This study not only demonstrates that neonatal FV and ICV treatments had the greatest impact on disease progression, it also helps disambiguate those two variables and demonstrates the independent significance of both age and route of administration. Our neonatal cohorts with both delivery methods (FV and ICV) were comparable to each other on most metrics in their improvement over the RO weaning-age cohort, suggesting an effect based primarily on age rather than route of delivery. The ability of systemic treatment in neonates to distribute more preferentially to the brain, resulting in higher hNPC1 copy numbers in the brain than RO mice, may be another key factor in explaining the efficacy of early FV treatment in conjunction with earlier mitigation of disease progression. One major advantage of neonatal treatment is that similar or strictly better outcomes can be achieved with significantly lower doses than our standard weaning-age systemic treatment. FV and ICV mice were given approximately 10-fold and 100-fold lower absolute doses, respectively, than RO mice with improved overall outcomes. As cost of treatment and AAV production capacities can be significant limitations in the clinic in terms of both cost and commercial production capacity, the ability to achieve greater efficacies with lower doses is vital to the success of clinically viable therapies and demonstrates the value of early screening and diagnosis [34–37].

Cerebrum copy numbers taken collectively correlate well with survival data; while not all other metrics correlate as clearly with cerebrum CNV, this suggests that a primary contributing factor to efficacy is the amount of vector successfully reaching the brain. One clear contributing factor to our FV cohort's overall performance was the amount of vector that reached the brain. Despite administering FV mice a dose of vector 10-fold lower than RO mice, FV mice still had as much or more total vector in the brain than RO mice. While RO delivery in our study showed similar brain:liver efficiency as other studies have shown in WT mice [38], advancing the timing of systemic delivery earlier alone was capable of improving brain delivery efficiency by approximately 200-fold. This substantial increase in proportional trafficking of systemically delivered AAV to the brain in younger mice may thus be part of the reason for improved efficacy in neonatal delivery. Biodistribution of AAV after injection is known to vary across tissues as a function of age of injection due to cells reaching post-mitotic stage in various tissues, such as the brain; different organ sizes and rates of organ growth during development; and differential vascular access during development [39]. Earlier systemic delivery with AAV9 thus showed a substantial advantage in efficiency of brain delivery while retaining potentially significant delivery to the periphery, though clinical translation specifically of the P0/P1 time point would depend on improved newborn screening, as discussed below. While capsid engineering is one way to improve efficiency to a target tissue relative to other tissues [40–46], this suggests that timing with respect to developmental stage can also directly improve efficiency. Nonetheless, higher gene copy in the brain is not sufficient to explain the improvement. The subpopulation of our FV cohort with low hNPC1 copy numbers in the cerebrum, comparable to that of our RO cohort, still survived longer than RO mice. This demonstrates that early treatment may be a contributing factor to efficacy for reasons beyond improved proportional delivery efficiency to the brain.

We found the most significant improvement in our neonatal cohorts by two quantitative metrics—weight loss progression and balance beam foot slips. The progression of the weight loss phenotype occurs in two distinct phases: initial weight gain until reaching a peak weight, and subsequent weight loss culminating in humane end point. A therapy that increases lifespan might prolong the first phase, slow the second phase, or improve both (i.e., delay the age at which weight loss begins and also slow eventual decline). The RO cohort reached peak weight at the same time as

saline-injected *Npc1^-/-* mice but specifically experienced a slowed and prolonged weight loss phase. Notably, both neonatal FV and ICV treatments extended the first phase of weight gain as well as the second phase of weight loss following peak weight. It thus seems that treatments given earlier with the opportunity to take effect earlier can prolong weight gain and delay symptom progression, allowing mice to continue growing before beginning their weight decline, while those treated later only experience the benefit in their weight phenotype in the decline phase. The balance beam test paints a positive picture for all treated groups. All treated groups have improved performance on the balance beam test at 9 wk, indicating that even with a later treatment, mice are seeing some near-immediate phenotypic benefit, including mice that saw no early benefit in their weight phenotype. The early cohorts provide even more impressive results, all showing a marked improvement over RO treatment. While ICV mice perform better than FV mice as a cohort, the high-CNV FV subpopulation performed comparably to ICV mice, demonstrating the capability of effective FV delivery to match ICV administration. As with the survival data, even the subpopulation of FV mice with CNV similar to that of RO mice completed the balance beam test better than RO mice, with 100% of low-CNV mice completing the test at 9 wk compared with only 77% of RO mice. This provides further evidence for the value of early treatment beyond simply a higher effective dose.

We also found no difference in efficacy between our ubiquitous EF1α(s) promoter and the truncated *Mecp2*-derived 546 promoter. Some truncations of the murine *Mecp2* promoter have been shown to drive restricted neuronal expression [47]. Previous work comparing different promoters in NPC1 gene therapy, such as comparing the EF1α(s) promoter with the neuron-specific CamKII promoter delivered by RO injection, did show improved efficacy with the EF1α(s) promoter [20], but the current study consistently showed the primary contributors to differences between the cohorts to be delivery route and age rather than promoter. Indeed, while liver involvement has also been described in NPC mouse models [48–50] and hepatomegaly is one of the most common early clinical signs in NPC individuals [3,51], hNPC1 copy number data in liver samples showed no correlation with survival or behavioral phenotype regardless of promoter, while CNS copy number does. Future studies will be needed to assess the qualitative and quantitative contribution of peripheral treatment to disease progression more directly in comparison with the more well-established contributions of CNS treatment.

Immunofluorescent imaging also shows the efficacy of both of our neonatal treatments, FV and ICV, in protecting key Purkinje cells in the cerebellum. Cerebellar Purkinje cell loss in NPC mouse models is one of the most notable phenotypes in the CNS [52]. RO treatments have previously been shown to delay, but not fully prevent, Purkinje cell loss, and in doing so, to slow the onset and progression of disease phenotypes and extend overall lifespan, with slower Purkinje cell loss correlating with disease improvements [20,21]. Despite this correlation, the near-complete preservation of Purkinje cells with ICV treatment of *Npc1^-/-* mice has been previously demonstrated without resulting in complete rescue from lethality [23]. End-stage imaging of brains revealed the potential for long-term Purkinje cell preservation as well, with our ICV cohort retaining nearly a full complement of Purkinje cells while still ultimately reaching end-stage milestones due to weight loss. Follow-up studies to this work would benefit from quantification of Purkinje cell survival, a limitation of this current project.

Notably, we also demonstrated for the first time with our neonatal FV cohort that an early systemic treatment can preserve the vast majority of Purkinje cells through end stage. Biodistribution studies have previously shown that, in contrast to adult intravenous injection, neonatal intravenous injection of AAV9 can transduce cerebellar Purkinje cells with high efficiency [53]; in a subset of mice we treated with this delivery method, we successfully rescued nearly all Purkinje cells. No systemically delivered treatments of NPC mouse models published to date have achieved such profound impact on the Purkinje cell survival. While neonatal mice have more immature blood-brain barrier (BBB) development than neonatal humans, the potential value of this delivery method, especially with the development of novel engineered AAV capsids with improved blood-brain barrier-penetrating capabilities [40–44], remains promising. Further, our small group of FV mice with preserved Purkinje cells trended toward not only significantly longer lifespans than FV mice with unpreserved Purkinje cells, but even longer lifespans than ICV mice. This trend may validate the importance of treatment of the periphery in addition to the CNS for optimal benefit. Efforts to improve delivery efficiency to the brain by capsid reengineering often

focus on liver detargeting [45]; for NPC, the potential improvement with systemic treatment suggests that retaining delivery to the liver may be important. Liver copy numbers were higher in RO mice than FV mice, likely due to a combination of later administration, greater total dose, and greater trafficking to the liver, though as noted above, this did not correlate with greater survival between the two systemically treated groups. To fully disambiguate the impact of age of delivery vs. delivery route, a comparison with weaning-age ICV-delivered treatment would be valuable. While neonatal delivery is not currently clinically feasible in the absence of more widespread newborn screening, systemic treatments also have significant practical advantages in human patients, even though they generally require higher doses of vector than targeted ICV delivery.

The Purkinje cell preservation observed in FV and ICV mice depicts mice at humane endpoint, much older than the saline-injected mice. This partly correlated with lifespan: some of the mice that retained their Purkinje cells had significantly extended lifespans, but others lived only as long as RO mice that suffered significant Purkinje cell loss. While previous data may have suggested that Purkinje cell rescue was the primary mediator of survival, and protecting Purkinje cells does seems to be necessary for rescuing $Npc1^{-/-}$ mice, the subset in which Purkinje cell rescue was insufficient to extend lifespan relative to those without significant rescue, both ICV and FV, points toward the potential existence of additional important targets in increasing lifespan. Further studies should attempt to identify additional specific populations of cells that can improve outcomes if protected along with Purkinje cells, including other CNS targets as well as peripheral organs.

In addition to Purkinje cells, imaging revealed less brain pathology in our ICV cohort. While GFAP staining for astrocytosis showed little demonstrable improvement in most cohorts, ICV mice showed an overall reduction of GFAP+ regions, with the areas of strongest reduction corresponding to the previously demonstrated biodistribution of ICV-injected AAV in neonates [54]. Improvement increased with higher delivered copy numbers as well. RO studies have shown minimal reductions in pathology in the brain [21]; while ICV treatment has previously been shown to improve astrocytosis in the cerebellum [23], this provides evidence that such treatments can reduce or prevent pathology in other regions of the brain as well and thereby protect key cell populations.

One limitation of our study is in comparing doses for each treatment. For systemic treatments to both weaning-age mice and neonates, we administered comparable doses of vector by weight of the mouse at the time of administration of approximately $1.0 \times 10^{14}$ GC/kg, in concert with FDA-approved dosages for other systemic AAV treatments, such as onasemnogene abeparvovec-xioi (ZOLGENSMA®, Novartis) for spinal muscular atrophy (SMA) ($1.1 \times 10^{14}$ GC/kg) [55] and delandistrogene moxeparvovec-rokl (ELEVIDYS®, Sarepta Therapeutics) for Duchenne muscular dystrophy ($1.33 \times 10^{14}$ GC/kg) [56]. This dose limitation is important to prevent toxicity from AAV administration [57–59], though it does lead to challenges in comparing data between groups. Adjusting to a consistent dose per body weight as we did for both neonatal and weaning-age systemic mice at time of injection rather than a consistent absolute dose leaves neonatally injected mice with fewer total copies of the vector per mouse than mice injected at an older age (Fig 1A, C). Further, many cell types that are largely post-mitotic in weaning-age mice are still dividing in neonates, leading to dilution of vector genomes and a different final biodistribution as discussed above.

Comparison between systemic and ICV treatments, which differ even more drastically in form and access to target tissues, results in an even greater challenge for dosage comparison. Choosing an appropriate ICV dose can also vary meaningfully for different disease models. In preclinical studies, neonatal ICV doses of AAV9-based gene therapy for SMA in a mouse model saw no benefit at doses below $2.7 \times 10^{10}$ GC/mouse [60]. By contrast, in an NPC mouse model, Hughes et al. found that a neonatal ICV dose of AAV9-hNPC1 as low as $4.6 \times 10^9$ GC/mouse showed clear efficacy in ameliorating disease phenotypes [23]. In clinical trials, doses are assigned based on brain mass. Three trials have used three different doses: A clinical trial for ICV-delivered AAV9 to treat MPS II used $6.6 \times 10^{10}$ GC/g brain mass, a trial for ICV-delivered AAV9 to treat Canavan disease used $3.1 \times 10^{10}$ GC/g brain mass, and a clinical trial for Rett Syndrome used $8 \times 10^{11}$ GC/g brain mass. A trial arm for Rett Syndrome with a dose of $2.4 \times 10^{12}$ GC/g was halted due to a patient death. A neonatal mouse brain weighs approximately 7.3 mg, leaving our delivered dose of $1 \times 10^{10}$ GC/mouse

equivalent to $1.36 \times 10^{11}$ GC/g brain mass, in the middle of the range of current trials. Even so, neonatal delivery results in a delivered dose different from the final mass of the brain, further complicating selection of appropriate dose. Another consideration was ensuring results that would allow for comparisons that could meaningfully isolate the variable of delivery route from dose. Delivering an ICV dose matching the absolute dose of systemic treatments would be inappropriate for comparison as the accessible tissue differs vastly between delivery methods. Normalizing to a consistent copy number delivered to the brain would ignore the substantially different biodistributions of each treatment. Further, higher brain CNV can be achieved via ICV treatment; systemic doses with AAV9 high enough to deliver the same dose of AAV9 to the brain are greater than commonly used in clinical settings and risk toxicity, as described above; conversely, a lower ICV dose tailored to match the CNV of RO delivery would be a drastic underdose relative to common feasible doses for ICV delivery of AAV gene therapy. With all of these considerations, we selected an ICV dose of $1 \times 10^{10}$ GC/mouse ($1.36 \times 10^{11}$ GC/g brain mass) that was intermediate with respect to preclinical trial doses of Hughes et al., who tested a low dose of $4.6 \times 10^9$ GC/mouse and a high dose of $2.5 \times 10^{11}$ GC/mouse, intermediate with respect to clinical trial doses, which ranged between $3.1 \times 10^{10}$–$2.4 \times 10^{12}$ GC/g brain mass at delivery to evaluate in comparison with our standard systemic dose. Notably, even at a lower dose than the maximum dose of Hughes et al.'s study, our neonatal ICV treatment for NPC still performed exceptionally well compared with systemic treatment. For all of these reasons, we chose standard clinical doses for all systemic treatments and an intermediate dose for ICV treatments. In combination, these allow for useful comparisons that include sets of similar treatments with paired quantitative features (e.g., similar effective doses in RO vs. FV mice) as well as sets of similarly standardized but structurally different treatments (e.g., RO vs. ICV, FV vs. ICV).

We conclude that earlier administration of AAV gene therapy can substantially improve disease progression outcomes of NPC model mice with significant Purkinje cell rescue possible via systemic treatment in addition to ICV treatment in neonates, though treatments at weaning age still provide some benefit. We additionally show that other targets may need correction for far-long-term survival in addition to cerebellar Purkinje cell rescue, and a comparison of immune response across delivery methods and age of delivery would be a valuable complement to such a broader assessment of additional factors contributing to efficacy. From a clinical perspective, while our weaning-age cohort did not fare as well as our neonatal cohorts, most children today are diagnosed with NPC beyond infancy and into school age. Efforts to expand newborn genetic screening for NPC are underway [61], but it is currently far from universal, even on existent newborn genetic screening panel programs. Our weaning-age results showing clear efficacy across multiple metrics provide additional evidence that treatment in young children is still valuable right now in the absence of newborn diagnosis. Further, novel engineered AAV capsids with improved CNS transduction and blood-brain barrier-penetrating capabilities may enable improved treatment at later ages [40–44]. While other cell populations may indeed be necessary to target as well, rescuing Purkinje cells still provides drastic benefits to disease outcomes even in the absence of such. Our comparison of neonatal and weaning-age treatments for the first time demonstrates improvements with earlier administration of NPC gene therapies via systemic delivery in addition to direct ICV delivery, particularly when successfully reaching the cerebellum, and we show that the cerebellum can be rescued via both delivery avenues, pointing the way for improved future therapeutic approaches for NPC.

## Supporting information

**S1 Fig. Promoter choice has minimal impact on vector delivery, survival, or phenotypic outcomes in all groups.** Cohorts treated with vectors containing the EF1α(s) and those treated with vectors containing the 546 promoter were compared for each analysis. Groups were compared via two-way ANOVA with multiple comparisons corrected by Holm-Šídák method unless otherwise indicated. Unless otherwise specified, *$P < 0.05$, **$P < 0.01$, ns: not significant. (A) Copy number in the cerebrum. (B) Copy number in the liver. (C) Kaplan-Meier survival curves. Survival curves were compared pairwise via log-rank Mantel-Cox test with Bonferroni correction applied manually. $\alpha = 0.0167$ (D) Percent increase or

decrease in weight at 9 wk of age compared with weight at 6 wk. (E) Composite phenotype scores at 9 wk. (F) Balance beam foot slips at 9 wk of age.
(TIF)

**S1 Data. Phenotype measurements and copy number data.**
(XLSX)

## Acknowledgments

The authors thank Cristin D. Davidson, Charles P. Venditti, and Forbes D. Porter (National Institutes of Health) for discussions and feedback over the course of the study; Attilio Iemolo (Novartis) for histology expertise and assistance, discussions, and feedback over the course of the study and review of the manuscript; Pamela K. Burrows (National Institutes of Health Library) and Zvi H. Rosen (Florida Atlantic University, Department of Mathematics and Statistics) for assistance with statistical analyses; and Yolanda L. Jones (National Institutes of Health Library) for manuscript editing assistance.

## Author contributions

**Conceptualization:** Benjamin E. Epstein, Jonathan Flynn, William J. Pavan, Fatih Ozsolak.

**Data curation:** Benjamin E. Epstein, Gabrielle M. Soden, Avani Mylvara.

**Formal analysis:** Benjamin E. Epstein.

**Funding acquisition:** William J. Pavan, Fatih Ozsolak.

**Investigation:** Benjamin E. Epstein, Gabrielle M. Soden, Arturo A. Incao, Jonathan Flynn, Avani Mylvara.

**Methodology:** Benjamin E. Epstein, Arturo A. Incao.

**Project administration:** William J. Pavan, Fatih Ozsolak.

**Resources:** Jonathan Flynn, William J. Pavan, Fatih Ozsolak.

**Supervision:** William J. Pavan, Fatih Ozsolak.

**Visualization:** Benjamin E. Epstein, Avani Mylvara.

**Writing – original draft:** Benjamin E. Epstein.

**Writing – review & editing:** Benjamin E. Epstein, Gabrielle M. Soden, Jonathan Flynn, Avani Mylvara, William J. Pavan, Fatih Ozsolak.

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
