## [Decision Letter · Decision Letter 0]

5 Nov 2025

Dear Dr. Pavan,

Thank you for submitting your manuscript to PLOS ONE. After careful consideration, we feel that it has merit but does not fully meet PLOS ONE’s publication criteria as it currently stands. Therefore, we invite you to submit a revised version of the manuscript that addresses the points raised during the review process.

We look forward to receiving your revised manuscript.

Kind regards,

Kent Lai

Academic Editor

PLOS ONE

Journal Requirements:

2. Thank you for stating the following in the Competing Interests/Financial Disclosure * (delete as necessary) section:

This work was supported by the National Human Genome Research Institute's (NHGRI) Intramural Research Program (1ZIAHG000068-16) https://www.genome.gov/ (WJP). The funder had no role in study design, data collection and analysis, decision to publish, or preparation of the manuscript.  This work was also funded by an NIH Cooperative Research and Development Agreement (CRADA) with Novartis Gene Therapies, Inc. https://www.novartis.com/ (CRADA #C-073-2023_A1) (WJP, FO).

We note that you received funding from a commercial source: “Novartis Gene Therapies, Inc”

“I have read the journal's policy and the authors of this manuscript have the following competing interests: The NIH has filed patents on behalf of WJP on work related to NPC1 genes and the AAV gene therapy treatment of NPC1 (US Patent Publication Numbers 20180104289, 20210113635). FO and JF are at present, or were during the time of their contribution to this manuscript, employed by Novartis Pharma AG and received salaries and own stock in Novartis as part of their remuneration for employment; they have no competing interests as regards consultancies, patents or products in development or currently marketed. This does not alter the authors' adherence to all the PLoS ONE policies on sharing data and materials.”

Additional Editor Comments :

Dear Dr Pavan,

Your manuscript was reviewed by three experts in the field and me. There is overall enthusiasm for the work and therefore, I encourage you to improve the paper by fully addressing the comments of all reviewers. By the way, In Fig 2C, the number 6 was missing from 546. I look forward to your revised submission and if you need more time, please let me know.

Reviewers' comments:

Reviewer's Responses to Questions

**Comments to the Author**

1. Is the manuscript technically sound, and do the data support the conclusions?

Reviewer #1: Yes

Reviewer #2: Partly

Reviewer #3: Yes

2. Has the statistical analysis been performed appropriately and rigorously?

Reviewer #1: Yes

Reviewer #2: Yes

Reviewer #3: Yes

3. Have the authors made all data underlying the findings in their manuscript fully available?

Reviewer #1: No

Reviewer #2: Yes

Reviewer #3: Yes

4. Is the manuscript presented in an intelligible fashion and written in standard English?

Reviewer #1: Yes

Reviewer #2: Yes

Reviewer #3: Yes

Reviewer #1: The manuscript by Epstein et al. uses AAV-mediated gene therapy to treat NPC and compares several therapy regimens, including different promoters, various routes of administration, and two treatment ages. Although not directly translatable to the clinical setting due to the natural limitations of using an animal model, this is a relevant study for preclinical research and is scientifically solid.

The study concludes that there is no difference in phenotype improvement between a constitutive and a neuron-specific promoter; however, significant differences in phenotype improvement were observed depending on the route of administration (with ICV outperforming intravenous administration) and age (early treatment delays disease manifestations more efficiently).

The manuscript is well-written, and the discussion was broad and touched on all major points. However, I am curious to know if the authors have assessed more parameters besides phenotype improvement, such as toxicity and immune response in the different groups. These could be valuable and could significantly increase the impact of this study.

Regarding Figure 6: Have you quantified the images? Considering that many conclusions in the manuscript were drawn and based on the preservation of Purkinje cells, I strongly believe another assay and/or a quantitative measure should be performed to support these findings, and not rely solely on immunofluorescence staining.

Regarding data presentation and editing:

The text and the figures are not cohesive. In all figures, there are mismatches between the presentation and the description - for example, Figure 3D is mentioned in the text before 3B and 3C. Moreover, the legend in Figure 2 seems misplaced: there is no D in Figure 2, and no data on Purkinje cells in the figure. I suggest an extensive review of the figures and the results section to confirm they are aligned and coherent.

When presenting the data for high and low CNV for FV, have the authors considered splitting into 2 groups? It may improve both the understanding of the graphs and include a direct comparison between the groups. If kept as is, please include a legend for the intermediate gray dots in the figures. Though understandable, it is not mentioned anywhere what they represent.

Finally, I did not have access to the data, which should be immediately available as per the journal’s guidelines. Please provide the dataset as supplemental material or in a repository.

Reviewer #2: The manuscript evaluates how timing and administration route affect AAV-NPC1 efficacy, concluding importantly that early treatment and effective brain delivery are essential for improving disease outcomes. The results are novel and relevant for advancing NPC gene therapy research. However, there are a series of major concerns/comments for the study. Importantly, the choice/justification of dose selection and evaluation of only single dose levels across administration routes make it challenging to definitively support the overall conclusion that systemic neonatal IV is comparable to ICV. Data provided in the manuscript support that neonatal AAV-NPC1 at a high and clinically relevant dose level (of approximately 1E14 vg/kg) is comparable to the single selected and evaluated ICV dose level, which based on brain weight extrapolation appears to be considerably lower than doses leveraged in current AAV9 ICV clinical trials for neurological indications.

Major Comments to address

• Recommend modification of manuscript title, as provided data does not definitively support the conclusion that neonatal IV is comparable to ICV, but rather compares 2 RoAs at single dose levels. Recommend for example: “Comparison of systemic neonatal and intracerebroventricular AAV9 gene therapy delivery demonstrating improved behavioral and phenotypic outcomes in a mouse model of NPC1”.

• Please justify why ICV delivery was not evaluated in weaning-aged mice to compare with IV administration. This comparison is particularly relevant for clinical translation, as neonatal (P0/1) mice have an immature BBB and are comparable to late gestation humans, potentially overestimating brain biodistribution compared to P3+ animals. Since newborn screening for NPC is unavailable, treatment evaluation in neonates is less informative for determining whether IV or ICV delivery is superior for NPC. As the study links improved outcomes to effective NPC1 brain delivery, comparing AAV-NPC1 copy numbers in the brain after IV and ICV administration post-neonatally—when the BBB is fully formed—would be highly valuable.

• State whether phenotype assessors were blinded to treatment; if not, provide justification. Clarify this in the methods section and on line 316.

• Revise Figure 1 title to indicate that earlier administration improves cerebrum:liver ratios, not specifically due to facial vein RoA.

• Since AAV escapes CSF after ICV delivery (as noted in line 223), why weren't liver copy numbers measured for ICV? Recommend including this analysis—it could clarify peripheral transduction alongside brain delivery efficiency. Were hepatosplenomegaly differences assessed across treatments?

• Figure 2 should present graphs and analyses comparing EFS and MECP2 promoters, not just describe them in the text.

• It appears that Figure 2 or its legend contains an error, as there is no panel labeled 2D, and panel 2B may be mislabeled. Please correct the references to Figure 2A–C in the paragraph beginning at line 263.

• Line 381 include description of statistical analysis of ICV vs FV for the balance beam.

• The connection between the sentence starting line 393 and the rest of the paragraph is not clear, suggest to add further clarification.

• Figure 6 does not specify whether the analyses are conducted with EFS, MECP2 promoter, or a combination of both. Differences between ubiquitous and neuronal-specific transgene expression could influence the effect on neuroinflammation. Clarification is needed by including images for both promoters or providing a rationale for presenting selected data. Consistent evaluation of staining at 9–10 weeks across groups would provide valuable information.

• Figure 6, due to the division of the FV group into high and low effectors, representative images for both groups should be included and clearly labeled. Currently, 6C may not accurately reflect an average representative image for FV.

• The size and quality of the submitted Figure 6 images do not support the statements in the paragraph beginning at line 406. Formal image analysis of either whole sagittal sections or specific brain regions is needed to validate these claims.

• Line 430 Agree with effect based primarily on age, however should provide further clarification that the effect may be a combination of early intervention to mitigate disease progression and better access to the brain specifically in neonatal mice with IV delivery.

• Line 454 Add clarification that the P0/1 neonatal IV intervention in this study is not representative of clinical early intervention as outlined above.

• Line 518 Can’t accurately make this conclusion due to lack of evaluation of peripheral biodistribution following ICV delivery vs IV.

• The sentence beginning on line 521 should be deleted or substantially revised. Neonatal P0/1 IV treatment does not correspond to a realistic human intervention timepoint; a comparison with juvenile or weaning-age subjects would be necessary to draw this conclusion.

• Line 575 Any further clarification to add on how this dose compares to extrapolated doses used in clinical evaluation of ICV AAV9 approaches in neurological disorders (e.g. Rett syndrome), given the selected 1E10 GC/mouse is dose is quite a conservative dose and IV selected dose is at the upper end of clinically used. Improved efficacy could feasibly be expected with higher ICV doses that when extrapolated to clinical doses are still feasible, which would change one of the fundamental conclusions of the manuscript.

Minor recommendations to address

• Line 225 suggest to specifically highlight higher liver copies in RO mice were due to later administration compared to neonatal FV mice.

• Line 406 no evidence of specific microglial infiltration is provided, should be updated to more appropriate microglial activation.

• Include reference to additional ICV evaluation of AAV-NPC1 by Hughes et al 2023 in line 54, 423, 587.

Reviewer #3: Comments

1. Page 14, Line 108 “For the adult systemic (RO) cohort: Mice aged 31–34 d were anesthetized via isoflurane inhalation.” Mice aged 31-34 d are not considered adult mice. They are still considered juvenile or adolescent.

2. Page 14, Line 117 “For neonatal injections, vectors were delivered to entire litters of mice at P1 and genotyping was performed at 2 wk.” What is the translational relevance of “neonatal injections” as most patients with NPC could not be diagnosed at neonatal?

3. Page 14, Line 122 “Mice were weighed weekly beginning at 6 wk of age and as frequently as daily as they progressed toward humane end point.” Are there any reasons that the authors started to weigh the mice at 6 wk of age?

4. Page 25, Line 340 The weight of the mice influences their performance in the balance beam test. The authors need to verify that variations in balance beam test results are not attributable to weight differences among the groups.

5. Page 26, Line 365 “Mice that did not successfully traverse the beam without falling were not included.” What factors contributed to the mice's inability to cross the beam without falling? Are there valid reasons to exclude these mice from the study?

Minor comments

1. Please ensure the reference format is consistent (e.g., refs. 21, 23, 29, 58).

**Do you want your identity to be public for this peer review?** For information about this choice, including consent withdrawal, please see our Privacy Policy

Reviewer #1: **Yes:** Edina Poletto

Reviewer #2: No

Reviewer #3: No

---

## [Author Response · Author response to Decision Letter 1]

8 Jan 2026

Reviewer #1: The manuscript by Epstein et al. uses AAV-mediated gene therapy to treat NPC and compares several therapy regimens, including different promoters, various routes of administration, and two treatment ages. Although not directly translatable to the clinical setting due to the natural limitations of using an animal model, this is a relevant study for preclinical research and is scientifically solid.

The study concludes that there is no difference in phenotype improvement between a constitutive and a neuron-specific promoter; however, significant differences in phenotype improvement were observed depending on the route of administration (with ICV outperforming intravenous administration) and age (early treatment delays disease manifestations more efficiently).

The manuscript is well-written, and the discussion was broad and touched on all major points. However, I am curious to know if the authors have assessed more parameters besides phenotype improvement, such as toxicity and immune response in the different groups. These could be valuable and could significantly increase the impact of this study.

We did not directly assess toxicity beyond the assessment that the treatments were generally well tolerated by the mice, and we did not assess immune response across groups. This would be a valuable future step, and the Discussion section has been amended to indicate this.

Regarding Figure 6: Have you quantified the images? Considering that many conclusions in the manuscript were drawn and based on the preservation of Purkinje cells, I strongly believe another assay and/or a quantitative measure should be performed to support these findings, and not rely solely on immunofluorescence staining.

We agree that quantification of Purkinje neurons would strengthen the finding of preservation in this important cell type. However, due to unforeseen circumstances, we no longer have tissues to carry out the suggested quantification. Furthermore, we do not have the financial resources, sufficient vector, nor the human power necessary to treat additional mice for tissue collection at an age-matched time point and subsequent quantification. As such, we have adjusted statements in the results to clearly indicate the subjective nature of this finding. We have also added a sentence in the discussion indicating that quantification of Purkinje neurons should be carried out in future studies and is a limitation of this current body of work.

Regarding data presentation and editing:

The text and the figures are not cohesive. In all figures, there are mismatches between the presentation and the description - for example, Figure 3D is mentioned in the text before 3B and 3C. Moreover, the legend in Figure 2 seems misplaced: there is no D in Figure 2, and no data on Purkinje cells in the figure. I suggest an extensive review of the figures and the results section to confirm they are aligned and coherent.

The caption for figure 2 and in-text references have been corrected to correspond properly.

The citation of figure 3D before 3B in the text is in accordance with PLOS figure guidelines (https://journals.plos.org/plosone/s/figures): “Lettered subparts of whole figures may be cited in any order in the text if the first mention of each whole figure is in numerical order. For example, you can cite any subpart of Fig 3 in any order (e.g., Fig 3C before Fig 3A), as long as Figs 1 and 2 have already been cited.”

When presenting the data for high and low CNV for FV, have the authors considered splitting into 2 groups? It may improve both the understanding of the graphs and include a direct comparison between the groups. If kept as is, please include a legend for the intermediate gray dots in the figures. Though understandable, it is not mentioned anywhere what they represent.

The data for high and low CNV has been separated out into distinct groups visually in Figure 2B–C and in the paragraphs describing the comparisons as well. A description of the grey dots is now included in the figure caption.

Finally, I did not have access to the data, which should be immediately available as per the journal’s guidelines. Please provide the dataset as supplemental material or in a repository.

The dataset has now been included as Supporting Data 1.

Reviewer #2

The manuscript evaluates how timing and administration route affect AAV-NPC1 efficacy, concluding importantly that early treatment and effective brain delivery are essential for improving disease outcomes. The results are novel and relevant for advancing NPC gene therapy research. However, there are a series of major concerns/comments for the study. Importantly, the choice/justification of dose selection and evaluation of only single dose levels across administration routes make it challenging to definitively support the overall conclusion that systemic neonatal IV is comparable to ICV. Data provided in the manuscript support that neonatal AAV-NPC1 at a high and clinically relevant dose level (of approximately 1E14 vg/kg) is comparable to the single selected and evaluated ICV dose level, which based on brain weight extrapolation appears to be considerably lower than doses leveraged in current AAV9 ICV clinical trials for neurological indications.

Major Comments to address

• Recommend modification of manuscript title, as provided data does not definitively support the conclusion that neonatal IV is comparable to ICV, but rather compares 2 RoAs at single dose levels. Recommend for example: “Comparison of systemic neonatal and intracerebroventricular AAV9 gene therapy delivery demonstrating improved behavioral and phenotypic outcomes in a mouse model of NPC1”.

The title has been updated in line with your suggestion.

• Please justify why ICV delivery was not evaluated in weaning-aged mice to compare with IV administration. This comparison is particularly relevant for clinical translation, as neonatal (P0/1) mice have an immature BBB and are comparable to late gestation humans, potentially overestimating brain biodistribution compared to P3+ animals. Since newborn screening for NPC is unavailable, treatment evaluation in neonates is less informative for determining whether IV or ICV delivery is superior for NPC. As the study links improved outcomes to effective NPC1 brain delivery, comparing AAV-NPC1 copy numbers in the brain after IV and ICV administration post-neonatally—when the BBB is fully formed—would be highly valuable.

The study was designed to focus on comparing earlier treatments to our own well established model of weaning-age systemic treatments, highlighting the importance of age of delivery and, with that in consideration, the most efficacious approach for implementing that. While it is true that BBB development differs in neonatal mice from even neonatal humans, we addressed the potential for this as an impact by highlighting comparisons between cohorts where total delivery to the brain was equivalent, such as in Fig 2 B–C, isolating delivery route by comparing FV-high mice and ICV mice with comparable levels of brain delivery and comparing FV-low mice to RO mice, also with comparable levels of brain delivery. Nonetheless, we agree that a comparison to weaning-age ICV treatment or even post-neonatal mice would be immensely valuable in the future to disambiguate the specific impact of age of delivery from BBB development stage. The Discussion has been updated to highlight these points.

• State whether phenotype assessors were blinded to treatment; if not, provide justification. Clarify this in the methods section and on line 316.

Phenotype assessors were blinded to treatment for assessments. This has been clarified in both the Methods section and on the line indicated.

• Revise Figure 1 title to indicate that earlier administration improves cerebrum:liver ratios, not specifically due to facial vein RoA.

The title for this figure has been clarified to more clearly specify the comparison in Fig 1E–G being highlighted.

• Since AAV escapes CSF after ICV delivery (as noted in line 223), why weren't liver copy numbers measured for ICV? Recommend including this analysis—it could clarify peripheral transduction alongside brain delivery efficiency. Were hepatosplenomegaly differences assessed across treatments?

Liver copies were measured for ICV mice as well and data is now included in figure 1C.

• Figure 2 should present graphs and analyses comparing EFS and MECP2 promoters, not just describe them in the text.

The two promoters are now compared in a new figure, Supporting Figure 1, for this and other metrics.

• It appears that Figure 2 or its legend contains an error, as there is no panel labeled 2D, and panel 2B may be mislabeled. Please correct the references to Figure 2A–C in the paragraph beginning at line 263.

The references in this paragraph and Figure 2 have been corrected.

• Line 381 include description of statistical analysis of ICV vs FV for the balance beam.

The statistical analysis methods for this are included in the Methods section:

“Slopes of linear regressions were compared pairwise via ANCOVA with Bonferonni correction applied manually for multiple comparison correction. … For statistical tests with Bonferroni correction applied manually, alphas for statistical significance of each comparison are listed along with P values.”

• The connection between the sentence starting line 393 and the rest of the paragraph is not clear, suggest to add further clarification.

The sentence has been reworded for clarification.

• Figure 6 does not specify whether the analyses are conducted with EFS, MECP2 promoter, or a combination of both. Differences between ubiquitous and neuronal-specific transgene expression could influence the effect on neuroinflammation. Clarification is needed by including images for both promoters or providing a rationale for presenting selected data. Consistent evaluation of staining at 9–10 weeks across groups would provide valuable information.

The title of figure 6 indicates that the images depict sections from mice injected with vectors containing the EF1α(s) promoter. A clarification that no differences were observed between mice in each promoter cohort has been added, consistent with our broader observations now shown in Supporting Figure 1 showing no difference between the two promoters.

While we agree that evaluation of images consistently at 9–10 weeks would provide additional valuable evidence, we no longer have the financial resources, sufficient vector, nor the human power necessary to treat additional mice for tissue collection at an age-matched time point. As such, we have adjusted statements in the results to clearly indicate the subjective nature of this finding. We have also added a sentence in the discussion indicating that quantification of Purkinje neurons should be carried out in future studies and is a limitation of this current body of work.

• Figure 6, due to the division of the FV group into high and low effectors, representative images for both groups should be included and clearly labeled. Currently, 6C may not accurately reflect an average representative image for FV.

The figure has been updated with clear labels, and an additional image of the FV-low group was added to the figure.

• The size and quality of the submitted Figure 6 images do not support the statements in the paragraph beginning at line 406. Formal image analysis of either whole sagittal sections or specific brain regions is needed to validate these claims.

We agree that quantification of images would strengthen the finding of preservation in this important cell type. However, due to unforeseen circumstances, we no longer have tissues to carry out the suggested quantification. As such, we have adjusted statements in the results to clearly indicate the subjective nature of this finding. We have also added a sentence in the discussion indicating that quantification of Purkinje neurons should be carried out in future studies and is a limitation of this current body of work.

• Line 430 Agree with effect based primarily on age, however should provide further clarification that the effect may be a combination of early intervention to mitigate disease progression and better access to the brain specifically in neonatal mice with IV delivery.

The effect explanation has now been clarified in greater detail.

• Line 454 Add clarification that the P0/1 neonatal IV intervention in this study is not representative of clinical early intervention as outlined above.

This clarification has been made in the manuscript.

• Line 518 Can’t accurately make this conclusion due to lack of evaluation of peripheral biodistribution following ICV delivery vs IV.

Biodistribution to the liver in ICV mice is now also included in figure 1C, providing justification for this explanation.

• The sentence beginning on line 521 should be deleted or substantially revised. Neonatal P0/1 IV treatment does not correspond to a realistic human intervention timepoint; a comparison with juvenile or weaning-age subjects would be necessary to draw this conclusion.

This sentence has been revised to focus on the practical advantages of systemic delivery more generally in the clinic.

• Line 575 Any further clarification to add on how this dose compares to extrapolated doses used in clinical evaluation of ICV AAV9 approaches in neurological disorders (e.g. Rett syndrome), given the selected 1E10 GC/mouse is dose is quite a conservative dose and IV selected dose is at the upper end of clinically used. Improved efficacy could feasibly be expected with higher ICV doses that when extrapolated to clinical doses are still feasible, which would change one of the fundamental conclusions of the manuscript.

The indicated paragraph has been updated with comparisons to doses used in Neurogene’s clinical trial for Rett Syndrome noted here as well as two other ICV AAV9 clinical trials – a trial from REGENXBIO for MPS II and a trial from Myrtelle for Canavan disease. The paragraph now notes the potential impact of higher doses and identifies the particular value of the comparison performed at this particular dose. Finally, the text now more explicitly communicates the goal of selecting doses to enable comparisons that permit isolating the variable of route of delivery from delivered dose.

Minor recommendations to address

• Line 225 suggest to specifically highlight higher liver copies in RO mice were due to later administration compared to neonatal FV mice.

This point has been added to the Discussion section.

• Line 406 no evidence of specific microglial infiltration is provided, should be updated to more appropriate microglial activation.

The text has been modified to specify microglial activation.

• Include reference to additional ICV evaluation of AAV-NPC1 by Hughes et al 2023 in line 54, 423, 587.

The text has been updated in the first two locations noted. This reference was not added to the final location as the dose used in Hughes et al 2023 was 1 × 1011 GC/mouse, and that segment of the Discussion specifically highlights minimal and maximal doses tested, which remain the range studied in Hughes et al 2021 as 4.6 × 109 GC/mouse and 2.5 × 1011 GC/mouse.

Reviewer #3

1. Page 14, Line 108 “For the adult systemic (RO) cohort: Mice aged 31–34 d were anesthetized via isoflurane inhalation.” Mice aged 31-34 d are not considered adult mice. They are still considered juvenile or adolescent.

This has been modified to “post-weaning-age (RO) cohort”.

2. Page 14, Line 117 “For neonatal injections, vectors were delivered to entire litters of mice at P1 and genotyping was performed at 2 wk.” What is the translational relevance of “neonatal injections” as most patients with NPC could not be diagnosed at neonatal?

The primary relevance of neonatal injections is to demonstrate that beyond the delimiter of pre-symptomatic vs. post-symptomatic, time of injection is a highly significant factor in the efficacy of treatment, and in this study, demonstrating that this is true not only for ICV delivery but also for systemic delivery. Additionally, as discussed already in the Discussion section, while most patients are diagnosed at a later age, efforts are underway to add NPC1 to newborn genetic screening panels.

---

## [Editor Report · Decision Letter 1]

18 Feb 2026

Comparison of neonatal systemic and intracerebroventricular AAV9 gene therapy delivery demonstrating improved behavioral and phenotypic outcomes in a mouse model of Niemann-Pick disease, type C1

PONE-D-25-44170R1

Dear Dr. Pavan,

We’re pleased to inform you that your manuscript has been judged scientifically suitable for publication and will be formally accepted for publication once it meets all outstanding technical requirements.

Kind regards,

Kent Lai

Academic Editor

PLOS One

Additional Editor Comments (optional):

The Editor found the revision satisfactory and moved to acceptance.
---

## [Editor Report · Acceptance letter]

PONE-D-25-44170R1

PLOS One

Dear Dr. Pavan,

I'm pleased to inform you that your manuscript has been deemed suitable for publication in PLOS One. Congratulations! Your manuscript is now being handed over to our production team.

Kind regards,

on behalf of

Dr. Kent Lai

Academic Editor

PLOS One